# Accelerating the Laboratory Testing Capacity through Saliva Pooling Prior to Direct RT-qPCR for SARS-CoV-2 Detection

**DOI:** 10.3390/diagnostics12123160

**Published:** 2022-12-14

**Authors:** Maria Mardalena Martini Kaisar, Sheila Jonnatan, Tria Asri Widowati, Helen Kristin, Suraj Rajan Vasandani, Caroline Mahendra, Soegianto Ali

**Affiliations:** 1Master in Biomedicine Study Program, School of Medicine and Health Sciences, Atma Jaya Catholic University of Indonesia, Jakarta 14440, Indonesia; 2Department of Parasitology, School of Medicine and Health Sciences, Atma Jaya Catholic University of Indonesia, Jakarta 14440, Indonesia; 3Nalagenetics Pte Ltd., Singapore 169204, Singapore

**Keywords:** pooling, saliva, SARS-CoV-2, direct RT-qPCR, COVID-19

## Abstract

The testing capacity of the laboratory is paramount for better control of the pandemic caused by SARS-CoV-2. The pooling method is promising to increase testing capacity, and the use of direct NAAT-based detection of SARS-CoV-2 on a non-invasive specimen such as saliva will ultimately accelerate the testing capacity. This study aims to validate the pooling-of-four method to quadruple the testing capacity using RNA-extraction-free saliva specimens. In addition, we intend to investigate the preferable stage of pooling, including pre- or post-heating. The compatibility of this approach was also tested on five commercial kits. Saliva specimens stored at −80 °C for several months were proven viable and were used for various tests in this study. Our findings revealed that pooling-of-four specimens had an overall agreement rate of 98.18% with their individual testing. Moreover, we proved that the pooling procedure could be conducted either pre- or post-heating, with no discordance and no significant difference in Ct values generated. When compared to other commercial detection kits, it demonstrated an overall agreement greater than 85%, which exhibits broad compatibility and ensures easy adaptability in clinical settings. This method has been proven reliable and increases the testing capacity up to fourfold.

## 1. Introduction

In the wake of nearly three years of the COVID-19 global pandemic, the capability to detect Severe Acute Respiratory Syndrome CoronaVirus-2 (SARS-CoV-2) remains crucial for pandemic control. The virus that initially emerged in December 2019 in Wuhan, China, has rapidly spread and caused unprecedented outbreaks, endangering the global population’s health. By November 2022, there were 640,395,651 confirmed cases and 6,618,579 mortalities worldwide [1], and these numbers will continue to rise. This necessitates the increase in testing capabilities that are high throughput and reliable. As recommended by the WHO, each country should conduct at least one test per thousand people per week [2]. According to *Our World in Data*, many countries were struggling to increase their testing rates at the beginning of the pandemic, thus it was difficult to handle the outbreaks. It was only in the second year that the testing rates increased significantly, allowing countries to have better control of the pandemic [3]. In order to avoid a future positivity surge regarding the ever-mutating virus, a developed testing method that allows an increased testing rate would be substantial. The gold standard for COVID-19 testing is the reverse transcription quantitative polymerase chain reaction (RT-qPCR), which is performed predominantly on nasopharyngeal and oropharyngeal (NPOP) swab specimens. However, this has certain limitations, such as requiring trained personnel for specimen collection, lengthy procedures, and expensive reagents [4]. Saliva is a clinically approved specimen for emergency use by the Food and Drug Administration (FDA) for SARS-CoV-2 diagnosis. The FDA allows for the self-collection of saliva; aside from being cheaper, it reduces the risk of infecting front-line healthcare workers; making saliva an excellent alternative specimen [5]. Previous studies have proven that heat-treated saliva as a direct RT-qPCR specimen bypasses the need for saliva specimens to undergo RNA extraction, thereby reducing cost and processing time while increasing testing capability [6,7,8,9]. The high agreement rates with NPOP specimens demonstrate the robustness of the developed method. Therefore, this straightforward method circumvents the limitations seen when extracting RNA, such as lengthy and costly procedures, thus can potentially increase the COVID-19 testing capacity. Distinct viruses causing other infectious diseases have been diagnosed using the pooling technique of various specimen types, such as hepatitis E virus detection using blood specimens and human cytomegalovirus using urine specimens [10,11]. The pooling method, otherwise known as the Dorfman method, allows us to increase testing capacity. This pooling method combines multiple specimens in a single test. In the condition of a negative pooling result, all samples are assumed to be below the test’s detection limit. In contrast, in a positively detected pool, each specimen is required to undergo further individual evaluation [12]. Evaluation has shown for SARS-CoV-2 detection that pooling of nasal and oropharyngeal swabs, along with saliva specimens, serves as a strategy to save reagents, increase the number of tests performed, and lower costs, particularly in regions with a low positivity rate of infection [13,14,15,16]. Other studies have demonstrated that pooled testing of NPOP swab specimens results in a 90–100% agreement on positive specimen detection, a comparable cycle threshold (Ct) value between pooled and individual specimens, and a slight decrease in sensitivity when individual specimens have a low viral load [13,17,18]. Although the implementation of pooling methods for SARS-CoV-2 detection on NPOP specimens and limited studies on saliva specimens have been carried out [16,17,19,20,21]. There are some essential aspects which have yet to be investigated, for example, at which stage should the pooling be carried out (before or after viral inactivation); how long should the heating be carried out, if this step is chosen as an inactivation step; and how is the utilization of this method in several commercial kits? Lastly, sensitivity can be altered by the number of pooled specimens, which is also affected by the prevalence of infections. Previous studies have proven that pooling more than five specimens could decrease the need for testing by more than 50% in countries with low prevalence. However, as prevalence increases, more pools will demonstrate positive results and require individual re-testing thereby increasing the cost [12,22]. Based on the remaining identified gaps, by choosing the moderate number of pooled specimens, we provide an improved pooling method of RNA-extraction-free saliva specimens to compare the approaches of pre-and post-heating and validate the method on five different commercial detection kits by utilizing the previously collected specimens.

## 2. Materials and Methods

### 2.1. Specimen Collection and Viability Test

A total of 52 saliva specimens were used in this study, consisting of five freshly collected specimens and 47 stored specimens obtained from the previous study [6]. As many as 27 positive and 20 negative stored saliva specimens were kept for approximately seven months at −80 °C in the COVID-19 Laboratory Center, School of Medicine and Health Sciences, Atma Jaya Catholic University of Indonesia (AJCUI).

Viability tests were performed on 47 specimens to determine the quality of the specimens’ post-storage. The evaluation used a direct RT-qPCR of the heat-treated saliva (at 95 °C for 10 min) method as developed by Mahendra et al. [6]. Specimens that exhibited concordance results with pre-storage testing were regarded as viable. Positive specimens were the ones with detected Ct values after long-term storage. Specimens that previously had no detected Ct values and remained undetected were considered negative specimens. Those viable specimens were selected for further evaluation.

### 2.2. RT-qPCR for SARS-CoV-2 Detection

XABT Multiple Real-Time PCR Kit for Detection of 2019-CoV (Beijing Applied Biological Technologies Co, Ltd, Beijing, China, Catalog #CT8223-48T) was used as the reference nucleic acid detection kit in this study. The RT-qPCR master mix was prepared according to the manufacturer’s instructions. Each reaction consists of 2 μL of nuclease-free water, 10 μL of the nucleic acid amplification reaction solution, 1 μL of reverse transcriptase, and 2 μL reaction of either solution A or B. Each master mix solution (A and B) was combined with 5 μL of pre-inactivated saliva specimen. Solution A detects the ORF1ab gene and the N gene, whereas solution B detects the E gene. Upon viral RNA template addition, strip tubes were briefly spun down to ensure that all liquid was positioned at the bottom of the tube. Thermocycling conditions were as follows: 45 °C for 10 min, 95 °C for 5 min, 45 cycles of 95 °C for 15 s, and 60 °C for 45 s—fluorescence was measured at 60 °C. Negative and positive controls were included in each RT-qPCR run. The amplification and detection of genes were performed using the CFX96 Touch Real-Time PCR detection system (Bio-Rad laboratories, Hercules, CA, USA). Cycle threshold (Ct) values were analyzed using CFX Maestro 1.0 software version 4.0 (Bio-Rad laboratories, Hercules, CA, USA). Amplification of internal control genes in specimens with no amplification (N/A) or Ct values greater than 40 on any of the target genes was interpreted as a negative result. Comparatively, in the pooling approach of four specimens, the detection of a single viral gene with a Ct value under 40 was regarded as positive and was then individually tested.

### 2.3. Validation of Pooling Strategy Pre- and Post-Heating

The overall workflow is provided in Appendix A. In total, ten negative saliva specimens were used, of which five were freshly collected and the other five were stored specimens. All specimens were spiked with the positive control from the reference detection kit. This process was carried out to compare the quality of the stored and freshly collected saliva specimens.

To obtain optimal results for both fresh and stored saliva specimens, the pooling strategy was evaluated using two approaches: (i) pre-and (ii) post-heating. To compare the performance of pooling approaches, Ct values were used as the parameter.

There were four distinct combinations of specimens in each pool, as each pool was designed to be a pool of four. Each specimen with a volume of 50 μL was spiked with 1 μL, 2 μL, 3 μL, 4 μL, and 5 μL positive control from the reference kit. In the first pooling approach, 50 μL of spiked negative saliva specimens were added to a specimen pool of 150 μL containing 50 μL each of three distinct negative specimens. The pool of four specimens was then heated to 95 °C for 10 min and directly added as 5 μL templates into the PCR reaction, which amounted to a volume of 20 μL. Whereas in the second pooling approach, 50 μL of spiked negative specimens and 50 μL of the other three negative specimens remained in separate microtubes. They were then heated separately at 95 °C for 10 min. Subsequently, four specimens were pooled into one microtube with a total volume of 20 μL subjected to PCR processing.

### 2.4. Pooling Approaches Using a Combination of Four Specimens

The stored saliva of 20 positive and 12 negative specimens assessed in the viability tests was used in this step. All specimens were tested using different approaches by combining specimens in a pool size of four, containing one positive (P4/1+), two positives (P4/2+), three positives (P4/3+), and all four positives (P4/4+) in each pool (Figure 1).

Each saliva specimen was pipetted at a total volume of 100 μL in a 1.5 mL microtube. All specimens were heated at 95 °C for 10 min and pooled post-heating. Due to some limited volume of specimens, the experiment was more beneficial to be carried out post-heating than pre-heating. Therefore, when there were positively detected pools, we could directly carry out the individual re-test instead of starting from the heating process.

### 2.5. Compatibility of RNA-Extraction-Free Saliva across Other SARS-CoV-2 PCR Detection Kits

We evaluated the reproducibility of the method using five different commercial detection kits. This step involved testing the compatibility of stored saliva specimens that had previously been evaluated for viability. New sets of pools were evaluated using XABT, Ardent Novel Coronavirus (COVID-19) Nucleic Acid Detection Kit (PCR-fluorescent probe), SD Biosensor Standard M nCoV Real-Time Detection Kit, BioSewoom Real-Q 2019-nCoV Detection Kit, and Tianlong SARS-CoV-2 Nucleic Acid Detection Kit (Real-time RT-PCR Method). A total of 39 pools for each kit were formed, including 15 pools of P4/1+, 13 pools of P4/2+, 6 pools of P4/3+, and 5 pools of P4/4+. The pools were compared with the individual PCR results of each specimen using the five kits accordingly. All procedures and interpretations were conducted following the manufacturer’s instructions. Specification of the kit details on targeted genes, the number of cycles and temperature, Ct value cut-off, master mix, and specimen volume are listed in Appendix A. The groups of pooled saliva were tested using the heating procedure prior to the RT-qPCR process, and the Ct difference (∆Ct) values were analyzed.

### 2.6. Data Management and Statistical Analysis

All collected data were stored in a Microsoft Excel database. Statistical analysis and visualization were performed using GraphPad Prism 9.1.1 (GraphPad Software, La Jolla, CA, USA). The results were reported as ∆Ct, determined as the pooled Ct minus individual Ct. A paired *t*-test was used to calculate the *p*-value to compare the Ct difference between pre-and post-heating (validation of pooling strategy pre- and post-heating) and between the pooled Ct value and individual Ct value (pooling approaches using a combination of four specimens). Differences with a *p*-value < 0.05 were considered to be statistically significant.

In addition, we calculated the descriptive statistics (mean and standard deviation) used to interpret the concordance between the dilution effect of the pooling method and the positive control (PC) and validate the pre-and post-heating pooling method. The compatibility of the developed method was reported as an overall agreement calculated using the following formula:$$\mathrm{Overall}\;\mathrm{agreement}\;(\%)=\left(\frac{\mathrm{true}\;\mathrm{positive}\;+\;\mathrm{true}\;\mathrm{negative}}{\mathrm{true}\;\mathrm{positive}\;+\;\mathrm{false}\;\mathrm{positive}\;+\;\mathrm{true}\;\mathrm{negative}\;+\;\mathrm{false}\;\mathrm{negative}}\right)\;\times \;100$$

## 3. Results

### 3.1. Saliva Specimens Stored at −80 °C Remained Viable

Pre-storage and post-storage testing were conducted using the DaAn Gene PCR (DaAn Gene, Guangzhou, China) and XABT detection kit (Beijing Applied Biological Technologies Co, Ltd., Beijing, China), respectively. The results of our tests indicated that amongst 47 specimens, 44 specimens were in concordance with the pre-storage testing and considered viable for further experiments. The discrepancy of the three specimens is that one specimen exhibited negative results and the other two specimens presented only a single gene signal; N gene with a Ct value of 35 and E gene with a Ct value of 37 were detected for each specimen. According to the manufacturer’s protocols, single detected gene signals were defined as negative, even though they were tested as positive seven months prior. They showed a 93.62% agreement between pre- and post-storage testing (Table 1). Pertaining to the Ct value, the comparison between the two detection kits (DaAn and XABT) cannot be directly made due to the difference in specificity and sensitivity. They can be influenced by the kits’ different limits of detection (LOD) (Appendix A). However, the Ct values from each specimen pre- and post-storage would provide the amount of available genetic material in those specimens as consideration (Figure 2). Additionally, the XABT kit (Beijing Applied Biological Technologies Co, Ltd., Beijing, China) targeted three genes, i.e., ORF1ab gene, N gene, and E gene. The Ct value range of the 24 positive specimens is displayed in Appendix A. Consequently, the results indicated that the 44 specimens were still viable and could be used for subsequent testing in this study.

### 3.2. Specimen Pooling Can Be Performed Both Pre- and Post-Heating

To maximize the use of limited available saliva SARS-CoV-2 specimens, the proof-of-concept experiment of pooling effect at two different stages, namely pre- and post-heating, was carried out using the synthetic viral-like material of SARS-CoV-2 provided as a positive control from the kit. This was intended to standardize the concentration between pools to interpret the effect of dilution attributed to the pooling method.

We spiked 1 μL, 2 μL, 3 μL, 4 μL, or 5 μL of positive controls from the reference PCR kit into the negative saliva for both stored and fresh saliva specimens. In Figure 3a,b, as expected, we found that the volume of positive controls affects the Ct value of spiked specimens. The resulting Ct values observed in saliva specimens tested were within the range of 25–35. Ct values decreased linearly proportional to the quantity of the positive control added. The same trends were observed in stored and fresh saliva specimens, which might indicate that long-term storage of saliva specimens did not affect the inhibition of the RT-qPCR process.

The negative specimens were spiked with the positive control and had various resulting Ct values (shown in Figure 3) became the model for the pooling validity test experiments carried out on pre- and post-heating. The varying volume of the added positive control did not cause any disparity in the stored and fresh saliva specimens’ results. At this stage of the study, there were also no missed detections. Next, we utilized freshly collected saliva specimens to observe the Ct values’ trends when pooling was carried out pre- and post-heating. The *p*-value of the targeted genes indicated no significant difference in the resulting Ct, as seen in Figure 4. Altogether, the results indicated that the pooling method could be performed pre- and post-heating.

### 3.3. Validation of Pooling of Four Approach for SARS-CoV-2 Detection from Saliva Specimens

For subsequent experiments, we used pooling at post-heating. We first validated the pooling-of-four-specimens approach. A significant difference was observed when comparing Ct values obtained from pooling to those obtained from testing individual specimens (Figure 5). When there is only one positive sample in a pool, the pooled Ct will be higher than the individual Ct of positive specimens due to the effect of dilution in the pooling method. However, when there is more than one positive specimen in a pool, the pooled Ct will be between the lowest individual Ct value and the highest individual Ct value in the pool.

In this study, we discovered a 98.18% overall agreement among 55 pools when tested using the reference PCR detection kit. No missed detection occurred in the pool of P4/2+, P4/3+, and P4/4+. However, one pool in P4/1+ exhibited missed detection (Table 2). Overall, the pooling-of-four method is applicable for individuals with a varying range of Ct values.

### 3.4. Saliva Pooling Approach Is Applicable in a Number of SARS-CoV-2 Commercial Detection Kits

We evaluated the pooling approach of direct PCR toward SARS-CoV-2 using five different commercial detection kits. All tested kits generated an invalid rate of below 10%, indicating that this method has a high level of reproducibility. XABT and Tianlong (n = 38/39) exhibited an overall agreement above 95%, whereas, Ardent (n = 35/39), SD Biosensor (n = 34/39), and BioSewoom (n = 34/39) exhibited an overall agreement greater than 85% (Table 3). Overall, our findings indicated that this developed method applies to multiple commercial kits with varying characteristics (Appendix A).

## 4. Discussion

Monitoring the viability for the testing of the stored saliva specimens was an additional objective of this study. In order to preserve specimens that could be useful for future research, it is critical to understand the condition for proper storage. Prior research indicated that specimens harboring coronaviruses could be stored at −80 °C for several years. This would prevent further degradation of salivary molecules while also preventing contamination by bacterial growth [23]. Our investigation revealed that 93.62% of the specimens stored for seven months (post-storage) were in concordance with pre-storage testing. Despite the head-to-head comparison, saliva specimen Ct values from pre- and post-storage testing cannot be directly compared due to the different RT-qPCR detection kits used in the previous and current studies. Both measurements yielded Ct values that were comparable. Overall, it can be concluded that the results of saliva specimen testing remain reliable when appropriately stored at −80 °C [24].

However, three specimens were in discordance with pre-storage testing. Post-storage testing indicated negative results, although a positive result was generated in pre-storage testing. When traced back, those specimens had a Ct value greater than 35 during pre-storage testing. Previous studies have proven that long-term storage may cause discordances in specimens with high Ct values [25]. This may be due to the viruses’ instability, which makes them susceptible to degradation during long-term storage. This phenomenon may occur because the majority of patients who are nearly cured have high PCR Ct values, suggesting the presence of residual viral debris instead of replication-competent viruses. When Ct values exceed 33, attempts to culture viruses from upper respiratory specimens are largely unsuccessful [26]. Nevertheless, −80 °C is highly recommended for long-term storage compared to room temperature or 4 °C, which can only preserve specimens for several days [25,27].

Numerous studies have reported that the pooling of specimens is an advantageous diagnostic technique for a vast array of viruses [28,29]. This method allows for a significantly greater number of specimens to be tested and is more cost-effective than individual testing [30]. In this study, we utilized saliva specimens, which allows for a more convenient sampling process. We validated the pooling-of-four method toward heat-treated saliva specimens, an RNA-extraction-free method previously described by Mahendra et al. [6]. A study by Watkins et al. pooled 5, 10, and 20 saliva specimens prior to RNA extraction and RT-qPCR detection of SARS-CoV-2 and reported a decrease in sensitivity of 7.41%, 11.11%, and 14.81%, respectively [31]. Therefore, a higher sensitivity decrease is avoidable by choosing pooled-of-four saliva specimens. Moreover, as saliva specimens did not undergo the conventional RNA-extraction step, limiting the pool size to four would reduce the possibility of missed detection. This is consistent with a previous study by Abdalhamid et al. [32], which demonstrated that pooling of more than five specimens increases the necessity for individual re-testing and are, therefore, less efficient. Another study by Jeong et al. [33] indicated that pooling four specimens is optimal in countries with a higher prevalence rate.

This study resulted in a satisfactory outcome; we achieved a 98.18% overall agreement when compared to the current conventional individual testing. However, one (P4/1+) pool exhibited missed detection (Table 2). When re-tested individually, the one positive specimen in this pool had Ct values of 33.64, 33.08, and 34.00 for ORF1ab, N, and E genes, respectively, whereas other pools of positive specimens with Ct values within the same range did not demonstrate any missed detection. This could be due to the varying viscosity of the saliva specimen, which may have led to pipetting errors and contributed to this anomaly. Without adequate homogenization before processing, pipetting was difficult, leading to decreased sensitivity. Without adequate homogenization before processing, pipetting was difficult, leading to decreased sensitivity [7]. Some studies indicate that the addition of Proteinase K may help to reduce the viscosity of saliva specimens [8,34,35]. However, this is still equivocal, as other prior studies also indicated that the addition of Proteinase K did not improve the outcome of testing using saliva specimens [6,36]. Further study might be needed to find a solution to this problem. However, with a high overall agreement of the pooling-of-four method, it can be concluded that this method is highly applicable for SARS-CoV-2 detection using RNA-extraction-free saliva specimens. Prior to processing, however, it is recommended to properly vortex the specimens to omit a sensitivity decrement resulting from the low homogeneity of the saliva specimens [7,37].

Further investigation revealed that pooling could be performed both pre- and post-heating as no discordance was detected. The volume of added positive control did not generate any disparities in the detection for both stored and fresh saliva specimens. Moreover, the Ct values did not exhibit a significant difference whether pooling was done pre- or post-heating (Figure 4). The stage at which the pooling method is implemented depends on the preferences of the users. When pooling is carried out pre-heating, cost and processing time are reduced. However, the recommended application of the pooling method is when the positivity rate is below 12% [12]. On the contrary, when the positivity rate is above 12%, we would recommend doing the pooling post-heating. Therefore, when there are positively detected pools, we can immediately repeat the PCR process of the specimens individually without going back to the heating process again.

In addition, we also demonstrate the applicability of the pooling method described in this study with multiple commercial detection kits. Agreement rates obtained were greater than 85% for all tested kits compared to the individual specimens. The sensitivity of the detection kits used can have an impact on the results. This method, on the other hand, is quite adaptable and may be used in laboratories with a multitude of detection kits. However, the variation observed among the tested kits suggests that validation prior to implementing this method is highly recommended.

The benefit of this method outweighed the slightly decreased overall agreement of 98%. The less processing time and higher testing capacity given by this method have proven beneficial in controlling the pandemic. Research by Joachim et al. [12], which conducted tests on several educational facilities in Germany, proved that pooling contributes to controlling virus transmission in the population. They managed to carry out the detection process and achieve results on the same day as sampling. With faster results, infected people can be contained earlier, while tracing would be more straightforward to prevent future outbreaks.

Our study was conducted in one of the central laboratories, utilizing a set of retrospective specimens from the Indonesian population. Therefore, more studies with similar experimental settings in different geographical areas would be advantageous, which would verify the widely applicable nature of this method. Furthermore, future studies aimed at gaining a deeper understanding of the salivary molecules that might affect the viral infection mechanism might be beneficial in reducing testing discordances. Despite this, our research revealed an advantageous specimen pooling strategy.

## 5. Conclusions

In conclusion, the pooling-of-four method using RNA-extraction-free saliva specimens is a reliable technique for detecting SARS-CoV-2, with potential for a quadruple testing capacity and cost savings on RNA extraction reagents. This method is applicable across other commercial detection kits; however, validation procedures are recommended to be performed in advance. Depending on the current prevalence, the pooling procedure can be performed at any stage based on the user’s preference. Furthermore, saliva specimens should be stored at −80 °C and would still be viable to be tested after several months.

## Figures and Tables

**Figure 1 diagnostics-12-03160-f001:**
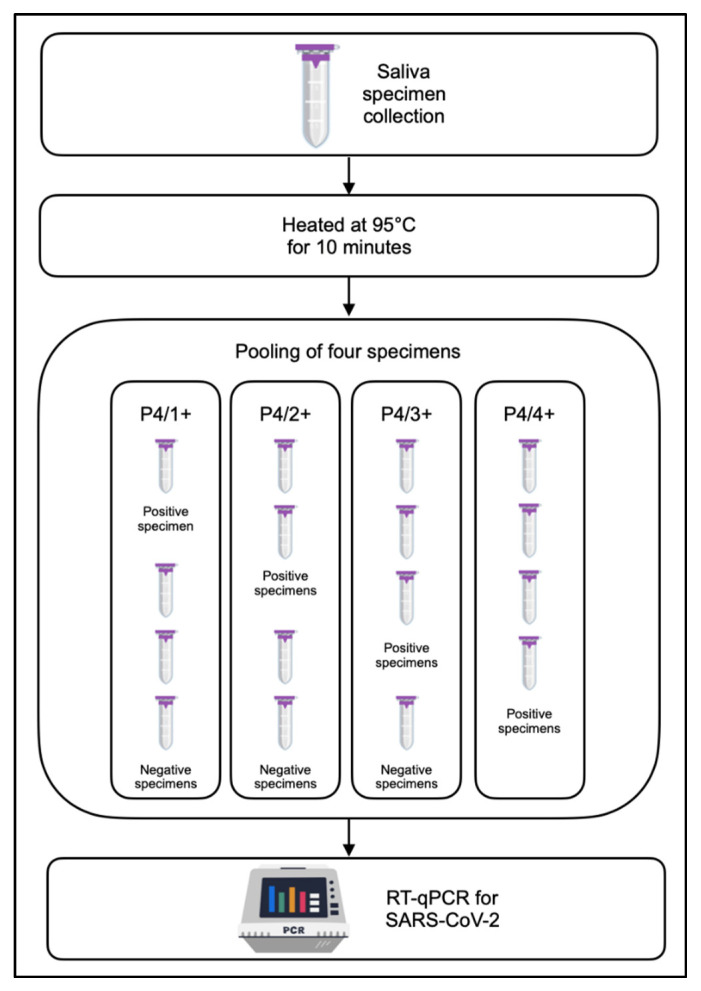
Schematic workflow of pooling of four specimens. Saliva specimens were collected and subjected to inactivation during heating. Three negative specimens and one positive specimen (P4/1+) were pooled, followed by RT-qPCR to detect SARS-CoV-2. The other pooling schemes are P4/2+, P4/3+, and P4/4+, which consist of two, three, and four positive saliva specimens, respectively, that were carried out accordingly prior to detection.

**Figure 2 diagnostics-12-03160-f002:**
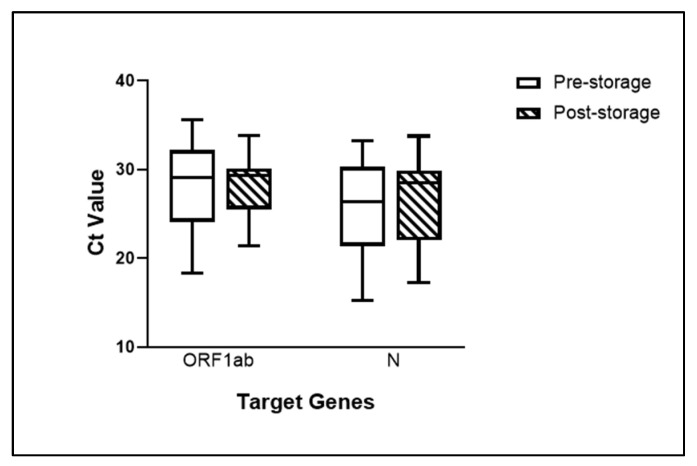
Specimens’ Ct values of targeted genes (ORF1ab and N genes) at pre- and post-storage. The white bars and shaded bars represented Ct value range of pre-storage specimens (detected by the DaAn Gene PCR kit (DaAn Gene, Guangzhou, China)) and post-storage specimens (detected by the XABT Multiple Real-Time PCR Kit of 2019-CoV (Beijing Applied Biological Technologies Co., Ltd., Beijing, China)). Results were interpreted by mean ± SD from 24 specimens.

**Figure 3 diagnostics-12-03160-f003:**
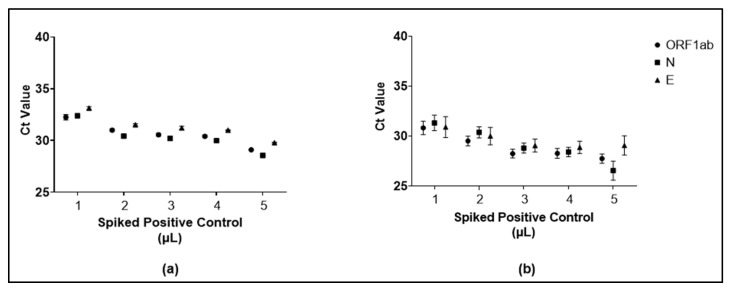
Ct value of the negative specimens with positive control spike-in. The difference in the resulting Ct value of ORF1ab, N, and E genes at each point of 1–5 µL spike-in of fresh saliva specimens (**a**) and stored saliva specimens (**b**). Results were interpreted by mean ± SD from triplicate testing.

**Figure 4 diagnostics-12-03160-f004:**
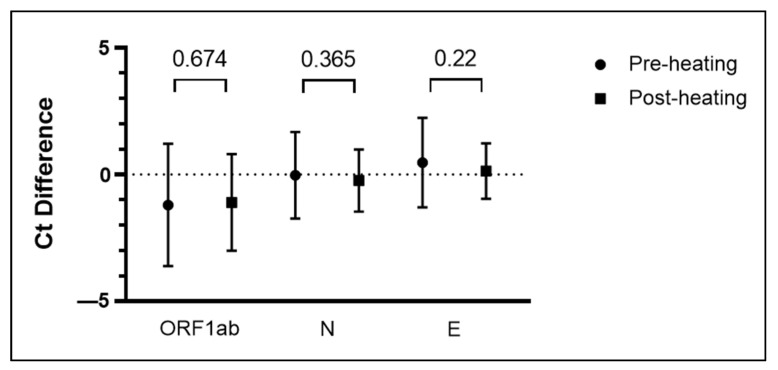
Validation of pre- or post-heating in pooling with positive control spike-in to negative saliva specimens, conducted on freshly collected saliva specimens. The paired *t*-test of Ct value difference between pre-and post-heating is indicated by the connecting line and *p*-value above it. Results were interpreted by mean ± SD from 18 pools of four with a range of 1–5 µL for positive control.

**Figure 5 diagnostics-12-03160-f005:**
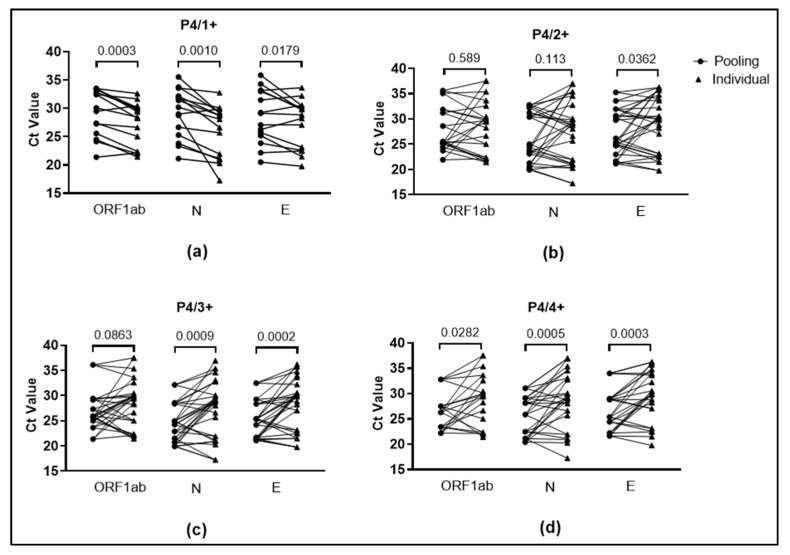
Comparison of pooled and individual specimens’ Ct values in the pooling-of-four method for each targeted gene. Ct value from 19 pools with one positive individual specimen (**a**); 16 pools with two positive individual specimens (**b**); 12 pools with three positive individual specimens; and (**c**); 7 pools with four positive individual specimens (**d**). Pooled Ct (black-colored circle) and corresponding individual Ct (black-colored triangle) are shown by connecting lines. *p*-value above plot graphs based on paired *t*-test.

**Table 1 diagnostics-12-03160-t001:** Viability test of stored specimens.

Post-Storage Testing	Pre-Storage Testing		Total
	(+)	(−)	
**(+)**	24	0	24
**(** **−)**	3	20	23
Total	27	20	47
Overall agreement		93.62%	

**Table 2 diagnostics-12-03160-t002:** Pooling-of-four method in different schemes.

Pools	Concordance to Individual Result (n)		Pools (n)	Overall Agreement (%)
	Concordant	Discordant		
P4/1+	19	1	20	95.00
P4/2+	16	0	16	100.00
P4/3+	12	0	12	100.00
P4/4+	7	0	7	100.00
Total	54	1	55	98.18

**Table 3 diagnostics-12-03160-t003:** Performance of commercial detection kits using pooled heat-treated saliva specimen as the RT-qPCR template.

Commercial Kit ^#^	Concordant to Individuals (n)	Discordant to Individuals (n)	Invalid Specimens (n)	Overall Agreement (%)
XABT *	38	1	0	97.43
Ardent	35	4	0	89.74
Tianlong	38	1	0	97.43
SD Biosensor	34	3	2	87.17
BioSewoom	34	5	0	87.17

* Reference kit ^#^ The testing conducted using 39 pools from stored saliva specimens (pools consist of: P4/1+ (15 pools), P4/2+ (13 pools), P4/3+ (6 pools) and P4/4+ (5 pools)).

## Data Availability

The dataset used and/or analyzed during the current study is available from the corresponding author on reasonable request.

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
