# Peer review of "Accelerating the Laboratory Testing Capacity through Saliva Pooling Prior to Direct RT-qPCR for SARS-CoV-2 Detection"

_diagnostics, 2022, doi:10.3390/diagnostics12123160_

Round 1

Reviewer 1 Report

The manuscript by Maria Mardalena Martini Kaisar et al described a saliva pooling method to increase the testing capacity of RT-qPCR SARS-CoV-2 detection. Related sample pooling strategies have been intensely investigated and more advanced methods haven been recommended with pooling sample size from 5-10. Thus, there is no much left in the significance of this manuscript. By the way, if the authors try to prove that the performance of their sample pooling method is better than others, they should make a detailed comparison.

Author Response

Dear Reviewer,

We would like to thank the reviewer for the positive comment and input for improving our manuscript on Pooling Saliva for SARS-CoV-2 RT-qPCR testing. We had made revisions to the manuscript accordingly.

  1. We rewrote some parts of the introduction and added some details of the references we used.
  2. In relation to the rewrite, we had put some relevant data from the cited references
  3. We had restated our conclusion to provide a clearer conclusion.
  4. Our proposed method of sample pooling is rather unique because we used direct RT-qPCR of the saliva bypassing the RNA extraction step. We also tested the method of pooling at the stage of pre and post-heating. We believe that our method could alleviate the burden of COVID-19 testing and screening, especially in countries with limited resources. 

Thank you very much for your input that helps us improve the manuscripts. 

Regards

Soegianto Ali.

Reviewer 2 Report

This study by Maria Mardalena Martini Kaisar et al., titled “Accelerating the Laboratory Testing Capacity through Saliva 2 Pooling Prior to Direct RT-qPCR for SARS-CoV-2 Detection”, aimed at validating the pooling-of-four method to quadruple the testing capacity using the RNA-extraction-free saliva specimens. The study also investigated the preferable stage of pooling, including pre- or post-heating. The outcome from this study provides an improved pooling method of RNA-extraction-free saliva specimens to compare the approaches of pre-and post-heating and validate the developed method on five different commercial detection kits by utilizing the previously collected specimens. Overall, the study is in good shape.

Comments:

·       The topic is relevant for the field as there is a need for increased testing capabilities that are high throughput reliable and cost effective, and this study that was conducted, is important in identifying these tests to detect Severe Acute Respiratory Syndrome CoronaVirus-2 (SARS-CoV-2) cases.

·       The experimental design used in the study is appropriate to test the hypothesis. They included a decent number of samples (52 saliva samples) in their study

·       The conclusions drawn are consistent with the evidence and arguments presented in this manuscript

Author Response

Dear Reviewer,

Thank you for your kind comment and input to our manuscript.

We believe that our proposed method is unique because we tested the pooling methods on saliva specimens that are directly used for RT-qPCR. Our proposed method could alleviate the burden of COVID-19 testing, especially in countries with limited resources.

We had made some revisions according to the inputs of both reviewers. Herewith we submit our revised version of the manuscript

Regards

Soegianto Ali

Round 2

Reviewer 1 Report

The revised manuscript has fit all my conerns.